# Study on Life Prediction Method of Ball Screw Base on Constructed Degradation Feature and IGWO-BiLSTM

**Qin Wu** [1,2,*]**, Jun Niu** [1] **and Xinglian Wang** [3]

1   College of Mechanical and Electrical Engineering, Lanzhou University of Technology, Lanzhou 730050, China;
    17355483732@163.com
2   Centre for Mechanical Efficiency and Performance Engineering, University of Huddersfield,
    Huddersfield HD1 3DH, UK
3   Electromechanical Instrument Operation and Maintenance Center, Lanzhou Petrochemical Company,
    Lanzhou 730060, China; wangxinlian@petrochina.com.cn
*   Correspondence: qwu.1973@hotmail.com

**Abstract:** Regarding the problem of reduced remaining useful life (RUL) due to wear of the ball screw in the feed system of CNC (computer numerical control) machine tools, a prediction method based on constructing the degradation feature vector of the signal data and the improved gray-wolf optimization with bidirectional long short-term memory (IGWO-BiLSTM) neural network regression model is proposed. Firstly, a time-domain analysis and the complete ensemble empirical mode decomposition with adaptive noise analysis (CEEMDAN) were carried out based on the collected life cycle signal data of a ball screw. The time-domain feature vector and the energy feature vector of each IMF (intrinsic mode function) component after CEEMDAN decomposition were constructed. The Pearson correlation coefficient was used to filter feature vectors and construct the multivariate feature vector. Secondly, this paper improves the traditional gray wolf optimization algorithm, adds a search strategy based on dimension learning, and combines the improved algorithm with the BiLSTM model, based on the IGWO-BiLSTM theory. A regression model between feature vectors and the remaining life of a ball-screw system was established. Finally, the prediction model was established according to the proposed method and compared with the other five neural network models: LSTM, BiLSTM, BO-LSTM (Bayesian optimization of LSTM), BO-BiLSTM, and IGWO-LSTM. The results indicate that this method has high accuracy and good generalization ability for predicting the remaining life of a ball-screw system.

**Keywords:** ball screw; CEEMDAN; IGWO; BiLSTM; life prediction





## 1. Introduction

CNC machine tools are widely used in modern industrial production, such as petrochemical, automotive, mold manufacturing, aerospace, and other fields. The development level of CNC machine tools reflects the overall advanced manufacturing technology level of a country. As an integral mechanical transmission component of the CNC machine tool feed system, the ball screw demands high precision, large bearing capacity, and reliability. The service life of ball screws may lead to mechanical failures and a decrease in production efficiency, indirectly affecting the machining accuracy of CNC machine tools and, thereby, affecting the economic benefits and image of the enterprise. In summary, it is necessary to establish an effective life-prediction model for ball screws to ensure the reliability of CNC machine tools [1,2].

In recent years, with the rapid development of Internet of Things technology, information technology, and artificial intelligence, predictive maintenance has gradually become a research hotspot [3–5]. Predictive maintenance mainly uses remaining useful life (RUL) prediction information to select the lowest-cost maintenance strategy and production schedule in the maintenance-opportunity window to reduce costs, improve efficiency, and maximize

production profits. In the context of predictive maintenance, researchers have carried out a series of life-cycle-operation tests of mechanical equipment. The Center for Intelligent Maintenance Systems at the University of Cincinnati has designed three sets of accelerated life-cycle fatigue tests of rolling bearings. The failure time of the bearing is determined by monitoring the wear debris in the oil. The data set clearly indicates the fault form after each bearing failure, so it can be used not only for residual life prediction research [6–8] but also for fault diagnosis research. The accelerated life test of rolling bearings was carried out at Xi'an Jiaotong University in China. The life cycle vibration signals of 15 rolling bearings under three working conditions were collected, and the failure parts of each bearing were clearly marked, which provided data support for the research in the field of health-status assessment [9,10]. The intelligent-learning model can independently learn the performance-degradation mode of mechanical equipment from the monitoring data through intelligent algorithms, and predict the remaining life. It does not need to construct a physical model or a statistical model in advance, so it has gradually become a research hotspot. At present, in the field of residual life prediction, the commonly used intelligent models mainly include the artificial neural network model, support vector machine model, and correlation vector machine model. The artificial neural network simulates the working process of the human brain through a large number of connection nodes in a complex hierarchical structure, which can automatically extract features from monitoring data and predict the remaining life of mechanical equipment. The support vector machine model is an intelligent model based on statistical learning theory, which can effectively deal with the residual life prediction problem of small sample data. The support vector regression machine model is also a common support vector machine model used in the field of residual life prediction. At present, some studies [11,12] have applied relevance vector machines to the field of residual life prediction. However, the prediction performance of relevance vector machines depends largely on the choice of the kernel function, and the optimization of model parameters is still a problem to be solved.

With the development of sensor technology and signal processing technology, the residual life prediction method based on deep learning of signal feature extraction has gradually become mainstream. Guo et al. [13] proposed the generalized variational mode decomposition (GVMD) algorithm to extract the weak features of rolling bearing faults. The GVMD algorithm can make full use of the bearing-fault frequency information and bandwidth information to accurately extract the weak feature components of bearing faults on demand. Zhang et al. [14] proposed a parallel variance constrained-convolutional auto-encoder (PVC-CAE) model for bearing degradation feature extraction. The PVC-CAE model is used to extract features in the frequency domain signal, and the LSTM network is used for prediction. In order to effectively evaluate the degradation trend of bearings, Qu et al. [15] proposed a prediction method of bearing RUL based on the beetle antennae BP (back propagation) neural network model optimized by the beetle antennae search algorithm. By extracting the time domain and frequency domain features of 18 kinds of bearing life-cycle vibration signals, the degradation features are constructed. Zhu et al. [16] proposed a rolling bearing fault feature enhancement extraction method based on the instantaneous angular speed (IAS) signal of the rotary encoder. The de-phasing algorithm (DPA) is used to suppress the strict periodic components, such as rotating frequency and its harmonics, and multi-point optimization minimum entropy deconvolution adjusted (MOMEDA) is used to enhance the rolling bearing fault impulse component. The spectrum analysis of the enhanced signal is carried out to extract the bearing fault impulse characteristics. Zhu et al. [17] proposed a new deep feature-learning method for RUL prediction through the representation of time-frequency (RTF) and a multiscale convolutional neural network (MSCNN), which improved the prediction accuracy. As a variant of the recurrent neural network (RNN), the long short-term memory (LSTM) network can effectively solve the problems of gradient disappearance and gradient explosion in RNN. Guo et al. [18] proposed a method based on empirical modal decomposition (EMD) and an LSTM network to predict the RUL of rolling bearings. However, LSTMs can only be passed from the

present moment to the future, which leads to ignoring the impact of the future on the present moment. At present, the research on the life prediction of ball screws based on data is few. Therefore, this paper chooses the BiLSTM model, which can consider both past and future information, to predict the RUL and improves the conventional parameter-seeking algorithm to highlight the advantages of the model.

Based on the above analysis, in order to accurately predict the remaining life of the ball screw, the authors improve the traditional gray wolf algorithm and add a search strategy based on dimension learning to enhance the balance between local search and global search. The improved algorithm is combined with a BiLSTM neural network to construct a new life-prediction regression model. Compared with the optimization results of five other regression models, such as LSTM, BiLSTM, etc., the method proposed in this paper performs the best in prediction accuracy and generalization ability through simulation data testing and verification.

## 2. Energy Characteristics of Signal IMF Component

### 2.1. CEEMDAN Algorithm

Torres [19] and colleagues proposed the CEEMDAN algorithm to solve the problem of modal aliasing in EMD (empirical mode decomposition). The EEMD (ensemble empirical mode decomposition) and CEEMD (complete ensemble empirical mode decomposition) algorithms, including Gaussian white noise, can improve mode aliasing, but they cannot effectively separate residual noise, and the added white noise is unevenly distributed in the high-frequency and low-frequency regions, leaving some white noise signals in the modal intrinsic components. To solve this issue, CEEMDAN adds adaptive white noise during each empirical mode decomposition, effectively addressing the problem of residual noise and incomplete decomposition that exist in EEMD and CEEMD [20–23].

The steps of the CEEMDAN algorithm are as follows:

Let $E_i(\cdot)$ be the i-th intrinsic mode function obtained by EMD decomposition, $IMF_k$ be the k-th intrinsic mode component obtained after CEEMDAN decomposition, $\varepsilon$ be the constant coefficient, and $\omega$ be the added random noise.

(1) The Gaussian white noise is added to the decomposed signal x(t) to obtain a new signal $x_i(t) = x(t) + \varepsilon_0 E_0(\omega_i)$ (i =1, 2, ..., N). The new signal is decomposed by the EMD algorithm to obtain the first-order intrinsic mode component $IMF_1$:

$$IMF_1(t) = \frac{1}{N} \sum_{i=1}^{N} IMF_1^i(t)_i \qquad (1)$$

Calculate the first margin:

$$r_1(t) = x(t) - IMF_1(t) \qquad (2)$$

(2) A new signal $r_1(t) + \varepsilon_1 E_1(\omega_i)$ (i = 1,2, ..., N) is obtained by adding noise to the residual component $r_1(t)$, and the second component $IMF_2$ is obtained by EMD decomposition:

$$IMF_2(t) = \frac{1}{N} \sum_{i=1}^{N} E_1(r_1(t) + \varepsilon_1 E_1(\omega_i)) \qquad (3)$$

(3) Calculate the margin:

$$r_k(t) = r_{(k-1)}(t) - IMF_k(t)(k = 2, 3, ..., K) \qquad (4)$$

(4) For the signal $r_k(t) + \varepsilon_k E_k(\omega_i)$ (i = 1, 2, ..., N), the k + 1th IMF is obtained by EMD decomposition:

$$IMF_{(k+1)}(t) = \frac{1}{N} E_k(r_k(t) + \varepsilon_k E_k(\omega_i)) \qquad (5)$$

(5) Repeat (3) and (4) until the margin is a monotonic function and cannot be further decomposed. The final residual sequence is $r_k(t) = x(t) - \sum_{k=1}^{K} IMF_k(t)$, then the original signal is decomposed into $x(t) = \sum_{k=1}^{K} IMF_k(t) + r_k(t)$.

The schematic diagram of CEEMDAN decomposition is shown in Figure 1:

Original Signal

Determine the amplitude ε and the average number N of the noise added

Add noise ω1 | Add noise ω2 | ... | Add noise ωN

EMD | EMD | ... | EMD

IMF1 | IMF2 | ... | IMFN

The mean value : $\mathrm{IMF}_1(t) = \frac{1}{N} \sum_{n=1}^{N} \mathrm{IMF}_1(t)_n$

Update residual : $r_1(t) = x(t) - \mathrm{IMF}_1(t)$

1st iteration

... The residual is processed above and iterated

Add noise ω1 | Add noise ω2 | ... | Add noise ωN

EMD | EMD | ... | EMD

IMF1 | IMF2 | ... | IMFN

The mean value : $\mathrm{IMF}_k(t) = \frac{1}{N} \sum_{n=1}^{N} \mathrm{IMF}_1(t)_n$

Update residual : $r_k(t) = x(t) - \mathrm{IMF}_k(t)$

*k*th iteration

Complete the decomposition：$x(t) = \sum_{k=1}^{K} IMF_k(t) + r_k(t)$

**Figure 1.** Schematic diagram of CEEMDAN decomposition.

*2.2. Signal IMF Component Energy Feature Construction*

When the ball screw is worn inside, the energy information of the IMF component decomposed by CEEMDAN will change accordingly. Here, the energy information of the IMF is used as the characteristics value to indirectly reflect the change of the RUL of the component [24].

The CEEMDAN method is used to decompose the signal x(t) to obtain n intrinsic mode functions (IMF components) and a residual r. The total energy of each IMF component is calculated, as shown in Formula (6):

$$E_i = \int_{-\infty}^{+\infty} | c_i(t) |^2 \, dt = \sum_{i=1}^{n} | C_i |^2 \tag{6}$$

In the formula, $C_i(t)$ is the i-th IMF component, $C_i$ is the amplitude of discrete points, and n is the number of sampling points. The total energy $E_s$ of m IMF components is:

$$E_s = \left( \sum_{i=1}^{m} | E_i |^2 \right)^{\frac{1}{2}} \tag{7}$$

Constructing the energy feature vector E′ are $E' = [E_1 E_2 \cdots E_m]/E_s$

## 3. IGWO-BiLSTM Regression Model

### 3.1. BiLSTM Neural Network

LSTM is a type of time-recurrent neural network that has evolved from the RNN (recurrent neural network). It was created to address the issue of "gradient disappearance" in the RNN structure of the recurrent neural network. The LSTM network improves upon the structure of the input layer, hidden layer, and output layer of the traditional RNN neural network. It includes a gate mechanism to control the path of information transmission, enabling selective memory or deletion of information that passes through the network.

Figure 2 shows a typical LSTM memory block structure. Each memory block has three 'gating' structures: the input gate, output gate, and forgetting gate.

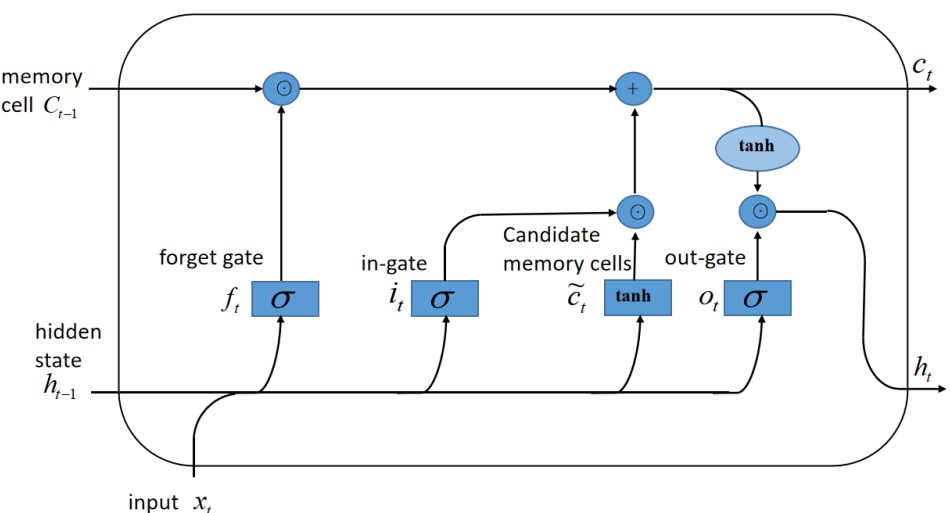

**Figure 2.** Schematic diagram of LSTM structure.

(1) The first step to calculating the forgetting gate $f_t$ is to determine the information discarded from the cell. The decision is implemented by the sigmoid layer of the forgetting gate. It looks at the previous output ht-1 as well as the current input $x_t$ and outputs a number between 0 and 1 for each number in a state $C_{t-1}$ on the cell, representing complete deletion and complete retention, respectively.

$$f_t = \sigma(W_f x_t + U_f h_{t-1} + b_f) \tag{8}$$

(2) The second step, the input gate $i_t$ determines what information is stored in the cell state next. The sigmoid layer of the input gate determines which values we will update. Next, the tanh layer creates a candidate vector $\widetilde{C}_t$, which will be added to the cell state. Combine these two vectors to create an updated value.

$$i_t = \sigma(W_i x_t + U_i h_{t-1} + b_i) \tag{9}$$

$$\widetilde{C}_t = \tan h(W_c x_t + U_c h_{t-1} + b_c) \tag{10}$$

(3) The third step to updating the previous state value, and update the previous state value $C_{t-1}$ to $C_t$.

$$C_t = f_t \otimes C_{t-1} + i_t \otimes \widetilde{C}_t \tag{11}$$

(4) The last step, the output gate $o_t$ needs to decide what to output. First, a sigmoid layer is run, which determines the part of the cell state to be output. Then the cell state is passed through tanh and multiplied by the output of the sigmoid gate. The output result $h_t$ is the output of LSTM and the hidden state of the next LSTM.

$$o_t = (W_o x_t + U_o h_{t-1} + b_o) \tag{12}$$

$$h_t = o_t \otimes \tanh(C_t) \tag{13}$$

where $W_i$, $W_f$, $W_o$, and $W_c$ represent the weight matrix from the input gate, forgetting gate, output gate, and candidate memory cells to the next input gate; $U_i$, $U_f$, $U_o$, and $U_c$ represent the weight matrix of the hidden layer; $b_i$, $b_f$, $b_o$, and $b_c$ represent the bias matrix of each gate structure, respectively. $\sigma$ is the sigmoid activation function.

BiLSTM (bidirectional long short-term memory) is an improvement of LSTM, and its structure diagram is shown in Figure 3. The BiLSTM neural network structure model is divided into two independent LSTM. The input sequence is input into the two LSTM neural networks in positive and reverse order, respectively, for feature extraction. The two output vectors, namely the extracted feature vectors, are spliced to form the final feature expression. In this model, the response signal of the ball screw inputs the information into the BiLSTM network layer through the input layer. The input sample signal outputs $\overrightarrow{h}_t$ through the forward LSTM layer and outputs $\overleftarrow{h}_t$ through the backward LSTM layer to jointly determine the value of the incoming hidden layer and obtain the output $y_t$ of BiLSTM. The update formula is as follows:

$$\overrightarrow{h}_t = LSTM(x_t, \overrightarrow{h}_{t-1}) \tag{14}$$

$$\overleftarrow{h}_t = LSTM(x_t, \overleftarrow{h}_{t-1}) \tag{15}$$

$$y_t = \overrightarrow{W}\overrightarrow{h}_t + \overleftarrow{W}\overleftarrow{h}_t + b_y \tag{16}$$

where $\overrightarrow{W}$ is the weight matrix from the forward LSTM to the output layer; $\overleftarrow{W}$ is the weight matrix from the inverse LSTM to the output layer; and $b_y$ represents the bias matrix of the output layer [25–28].

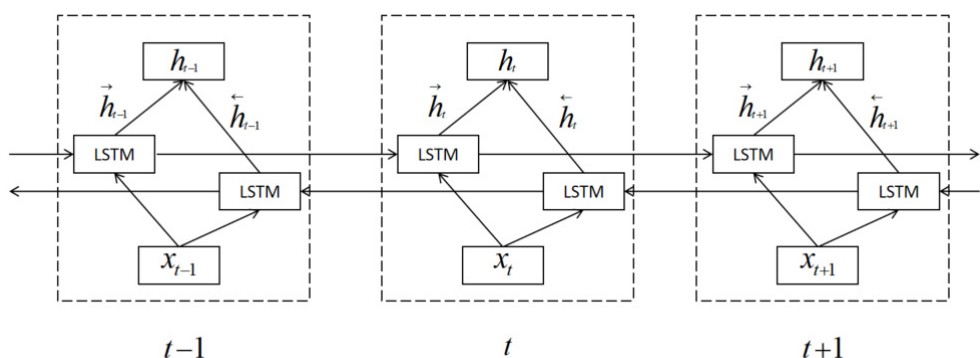

**Figure 3.** Schematic diagram of BiLSTM structure.

### *3.2. Gray Wolf Optimization Algorithm*

Gray wolf optimization (GWO) is a swarm intelligence optimization algorithm with a global optimal search mechanism. The inspiration comes from the predation behavior of the gray wolf group. There is a strict hierarchy in the gray wolf group, and a small number of gray wolves with absolute discourse power lead a group of gray wolves to prey. Gray wolves are generally divided into four levels: $\alpha$ wolves, $\beta$ wolves, $\delta$ wolves, and $\omega$ wolves. The rights are from large to small in order to simulate the leadership class, as shown in Figure 4. Collective hunting is a social behavior of gray wolves. Social hierarchy plays an important role in the process of collective hunting, and the process of predation is completed under the leadership of a wolf. It mainly includes three steps: (1) tracking, approaching, and harassing the prey; (2) hunting and encircling the prey until it stops moving; and (3) attacking the prey.

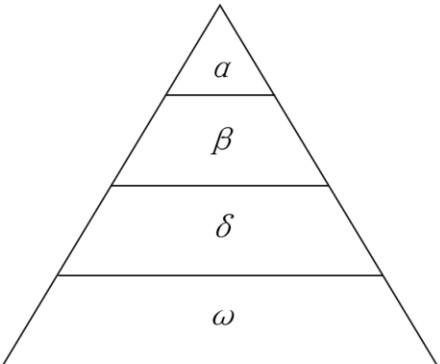

**Figure 4.** Gray wolf distribution level pyramid.

To begin, construct a gray wolf social hierarchy model and mathematically model the social hierarchy of the gray wolf. The $\alpha$ wolf is used as the optimal solution, that is, the individual's fitness is optimal, the suboptimal solution is $\beta$ wolf, and the best solution is $\delta$ wolf, which can be the global optimal solution or local optimal solution of the objective function, with the minimum objective function value or the maximum objective function value. The remaining candidate solution is named $\omega$ wolf. The hunting process is guided by the three wolves $\alpha$, $\beta$, and $\delta$, and the $\omega$ wolf follows the three wolves. That is, first find the three best solutions, and then search around the region, the purpose is to find a better solution, and then update the $\alpha$, $\beta$, and $\delta$ wolves. The behavior of gray wolves hunting prey is defined as follows:

The distance formula between individual and prey:

$$D = \mid C \cdot X_p(t) - X(t) \mid \tag{17}$$

Gray wolf position update formula:

$$X(t+1) = X_p(t) - A \cdot D \tag{18}$$

Coefficient vectors:

$$A = 2\alpha \cdot r_1 - \alpha \tag{19}$$

$$C = 2 \cdot r_2 \tag{20}$$

where t is the number of iterations, D is the distance vector between the individual and the hunt, $X_p$ is the position vector of the prey, X is the position vector of the gray wolf, a is the convergence factor (linearly decreases from 2 to 0 with the number of iterations), and $r_1$ and $r_2$ are random vectors, with modulo random numbers between 0–1.

Gray wolves can identify the position of prey and surround them. When the gray wolf recognizes the position of the prey, it guides the wolf group to surround the prey under the guidance of $\alpha$, $\beta$, and $\delta$. The mathematical model of gray wolf individual tracking prey position is described as follows:

$$\begin{aligned} D_\alpha &= \mid C_1 \cdot X_\alpha - X \mid \\ D_\beta &= \mid C_2 \cdot X_\beta - X \mid \\ D_\delta &= \mid C_3 \cdot X_\delta - X \mid \end{aligned} \tag{21}$$

Among them, $D_\alpha$, $D_\beta$, and $D_\delta$ represent $\alpha$ and $\beta$, respectively, and the distance from other individuals; $X_\alpha$, $X_\beta$, and $X_\delta$ represent the current position of $\alpha$, $\beta$, and $\delta$, respectively. $C_1$, $C_2$, and $C_3$ are random vectors, and X is the current position of the gray wolf.

$$\begin{aligned} X_1 &= X_\alpha - A_1 \cdot (D_\alpha) \\ X_2 &= X_\beta - A_2 \cdot (D_\beta) \\ X_3 &= X_s - A_3 \cdot (D_\delta) \end{aligned} \tag{22}$$

$$X_{t+1} = \frac{X_1 + X_2 + X_3}{3} \tag{23}$$

Formula (22) defines the step length and direction of ω individuals in the wolf pack towards α and β, and δ, respectively, and Formula (23) defines the final position of ω.

When the prey stops moving, the gray wolf completes the hunting process by attacking. In order to simulate approaching prey, the value of a is gradually reduced, so the fluctuation range of A is also reduced. In other words, in the iterative process, when the value of a decreases linearly from 2 to 0, the corresponding value of A also changes in the interval $[-a, a]$.

### 3.3. Improving Gray Wolf Optimization Algorithm

The classical GWO algorithm will cause the diversity loss of the wolf pack to converge prematurely so that the global optimal solution cannot be accurately obtained. The IGWO algorithm adds a dimension learning-based hunting (DLH) search strategy. The role of DLH is to enable each wolf in the group to learn from the surrounding wolves and exchange information, ensuring a fair selection of the best candidate wolves from the population, thereby enhancing the balance between local search and global search, ensuring the diversity of individuals and better searching of the entire space. It solves the shortcomings of the algorithm's poor optimization effect on complex problems. In the IGWO algorithm, three different steps are redefined, including wolf initialization, wolf movement, wolf selection, and update [29].

(1) Initialization stage. N wolves are randomly distributed and searched with $\left[l_i,\ u_j\right]$ in a given space.

$$X_{ij} = l_j + \text{rand}_j[0,1](u_j - l_j), i \in [1, N], j \in [1, D]_{\circ} \tag{24}$$

(2) Movement stage. The hunting strategy in IGWO is a combination of $X_{i\text{-GWO}}(t + 1)$ and $X_{i\text{-DLH}}(t + 1)$, that is, a combination of group-based hunting and dimension-based learning hunting (DLH). The Euclidean distance between the current position $X_i(t)$ and the updated $X_{i\text{-GWO}}(t + 1)$ is calculated. Equation (25) is used to construct the neighborhood $N_i(t)$ of each wolf.

$$R_i(t) = \parallel X_i(t) - X_{(i-\text{GWO})}(t+1) \parallel,$$
$$N_i(t) = \left\{ X_j(t) \mid D_i\big(X_i(t), X_j(t)\big) \le R_i(t), X_j(t) \in p \right\} \tag{25}$$

where the position of $X_j(t)$ is adjacent to the current iteration position $X_i(t)$; $D_i$ is Euclidean distance; and p is the population of gray wolves. The position of the candidate wolf is calculated by Equation (26), and the recommended update position is established.

$$X_{i-\text{GWO}}(t+1) = X_{i,d}(t) + \text{rand}(X_{n,d}(t+1) - X_{r,d}(t))_{\circ} \tag{26}$$

(3) Selection and update stage. By comparing the fitness values of candidate wolves $X_{i\text{-GWO}}(t + 1)$ and $X_{i\text{-DLH}}(t + 1)$, the best candidate wolves are selected, which can be described by Formula (27).

$$X_i(t+1) = \begin{cases} X_{i-\text{DLH}}(t+1), & f(X_{i-\text{GWO}}) < f(X_{i-\text{DLH}}) \\ X_{i-\text{GWO}}(t+1), & f(X_{i-\text{GWO}}) \ge f(X_{i-\text{DLH}}) \end{cases} \tag{27}$$

### 3.4. IGWO-BiLSTM Regression Model

The parameter determination of BiLSTM models is usually based on manual experience, which leads to a significant amount of time required for model parameter adjustment and a tendency to converge to locally optimal solutions [30]. The improved gray wolf optimization (IGWO) algorithm has the advantages of strong global search ability, fast convergence, and easy implementation. In order to enhance the accuracy of the residual useful life prediction model of a ball screw, the parameters of the BiLSTM model are optimized and adjusted using the IGWO. Specifically, the number of hidden layer units,

learning rate, and iteration times in the BiLSTM network are employed as the positions of wolves. By calculating the fitness function and updating the position of wolves, an optimal solution for the LSTM network model parameters can be obtained, and the residual life prediction model of the ball screw can be constructed using the optimal model parameters. The specific steps involved in constructing the life-prediction model LSTM are as follows:

Step 1: Divide the preprocessed sample data into a training set and a test set in the appropriate proportion.

Step 2: Specify the initial data for IGWO, including the number of gray wolf populations, initial coordinates, and iterations. Transform the number of hidden layer units, learning rate, and iteration times of the BiLSTM network into the position coordinates of wolves, then select the training sample set to train the BiLSTM model.

Step 3: Calculate each wolf's individual fitness value within the wolf group, and update their individual positions based on the fitness value calculated by the fitness function. Once either the maximum number of iterations is reached or the global optimal position satisfies the minimum boundary, the optimal solutions for the three model parameters of the hidden layer unit number, learning rate, and iteration number of the BiLSTM network will be obtained.

Step 4: Select the test sample set, and test the BiLSTM network using the above-optimized parameters to obtain the optimum BiLSTM network model.

## 4. Life Prediction Methods

The block diagram depicting the ball-screw life-prediction method based on CEEM-DAN decomposition and the IGWO-BiLSTM model is illustrated by MATLAB in Figure 5.

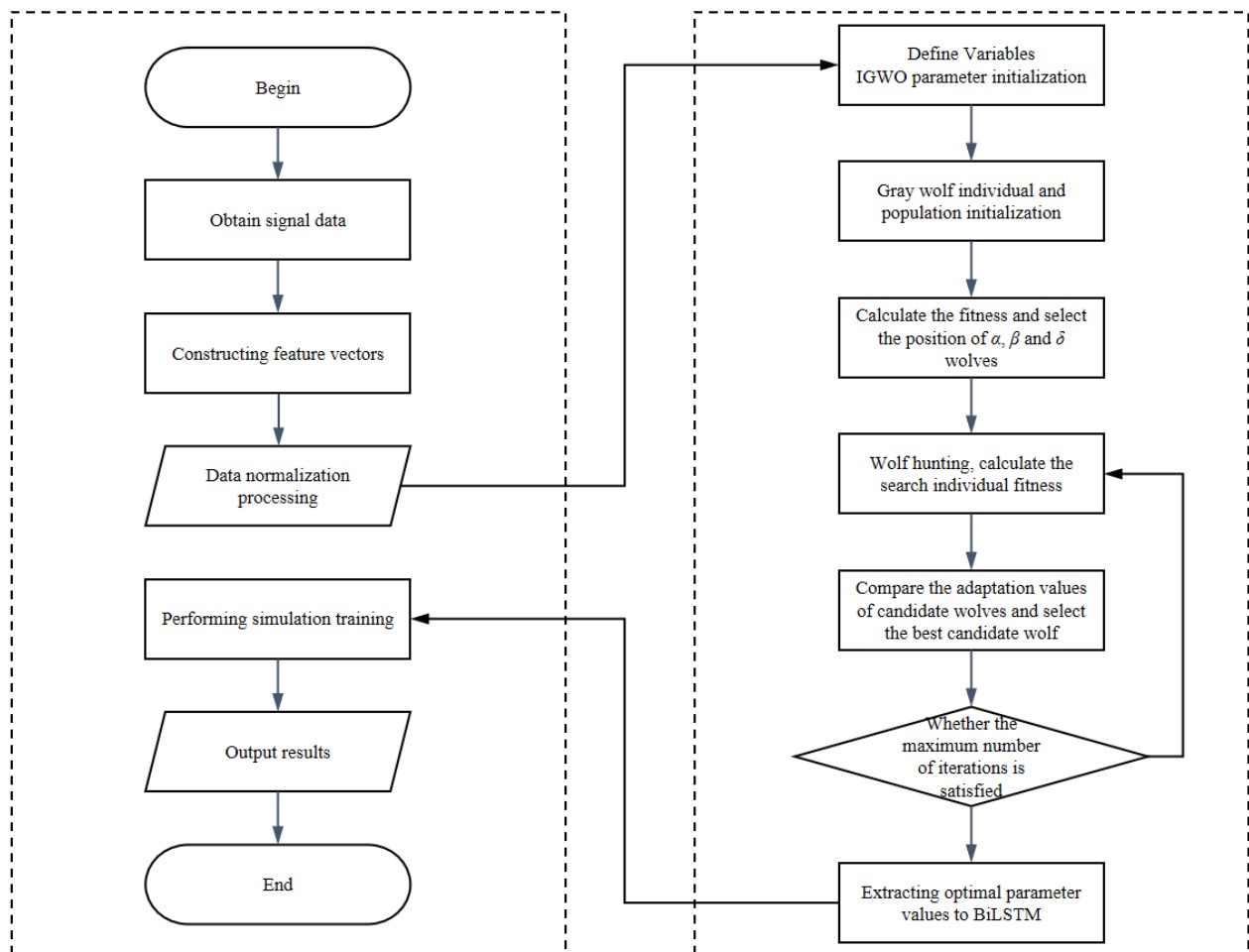

**Figure 5.** Block diagram of IGWO-BiLSTM ball-screw life-prediction method.

(1) Extract time-domain feature vectors. Extracting the time-domain characteristic values from the collected ball-screw drive-motor current signal, we have identified 16 types of time-domain characteristics. Their respective calculation formulas are provided in Table 1 [31].

**Table 1.** Equation for calculating time domain eigenvalues.

| Name | Formula |
|---|---|
| Maximum | $\max(x(n))$ |
| Peak | $\max \mid x(n) \mid$ |
| Mean | $\frac{1}{N} \sum_{n=1}^{N} x(n)$ |
| Root amplitude | $(\frac{1}{N} \sum_{n=1}^{N} \sqrt{\mid x(n) \mid})^2$ |
| Standard deviation | $\sqrt{\frac{1}{N-1} \sum_{n=1}^{N} [x(n) - \bar{x}]^2}$ |
| Kurtosis | $\frac{\sum_{n=1}^{N} [x(n) - \bar{x}]^4}{(N-1) \sigma_x^4}$ |
| Waveform factor | $\frac{\sqrt{\frac{1}{N} \sum_{n=1}^{N} x^2(n)}}{\frac{1}{N} \sum_{n=1}^{N} |x(n)|}$ |
| Pulse factor | $\frac{\max|x(n)|}{\frac{1}{N} \sum_{n=1}^{N} x(n)}$ |
| Minimum | $\min(x(n))$ |
| Peak value | $x_{max} - x_{min}$ |
| Absolute average | $\frac{1}{N} \sum_{n=1}^{N} \mid x(n) \mid$ |
| Variance | $\frac{1}{N} \sum_{n=1}^{N} x^2(n)$ |
| Root mean square | $\sqrt{\frac{1}{N} \sum_{n=1}^{N} x^2(n)}$ |
| Skewness | $\frac{\sum_{n=1}^{N} [x(n) - \bar{x}]^3}{(N-1) \sigma_x^3}$ |
| Peak factor | $\frac{x_{max}}{\sqrt{\frac{1}{N} \sum_{n=1}^{N} x^2(n)}}$ |
| Margin factor | $\frac{\max|x(n)|}{(\frac{1}{N} \sum_{n=1}^{N} \sqrt{|x(n)|})^2}$ |

Although the 16 time-domain features mentioned above can reflect various aspects of the performance status of parts, the accuracy of the reflection varies greatly. Certain features are closely linked to changes in the degradation state of parts, while others cannot accurately indicate trends in the change of parts' states. Therefore, it is necessary to select features from the set that can provide accurate reflections of performance state changes so as to avoid excessive dimension and ensure accuracy in predictions. In this case, the Pearson correlation coefficient ρ (Equation (28)) was used to represent the correlation between features and the component's performance, with a value range between [−1, 1].

$$\rho = \frac{\sum x_i y_i - n \bar{x} \bar{y}}{(n-1) s_x s_y} = \frac{n \sum x_i y_i - \sum x_i \sum y_i}{\sqrt{n \sum x_i^2 - (\sum x_i)^2} \sqrt{n \sum y_i^2 - (\sum y_i)^2}} \tag{28}$$

By applying Equation (28), the correlation coefficient between the 16 time-domain features listed in Table 1 and the RUL can be calculated by selecting the most relevant m eigenvalues to the residual life and constructing a time-domain feature vector $T = [T_1 \ T_2 \ \cdots T_m]$ as the input feature for the model.

(2) Extract energy characteristic values by decomposing the signal with CEEMDAN into several IMF components. Utilize Formula (28) to calculate the correlation coefficient between each IMF component and the original signal to obtain *n* components with a

strong correlation with the original signal, which contains the most crucial information. Following the approach in Section 2, use the energy information feature vector $E' = [E'_1 \, E'_2 \cdots E'_n]$ of these IMF components as the energy input feature for the model. Combine the obtained time domain feature vector with the energy feature vector to create a new feature vector, which serves as the input feature for the IGWO-BiLSTM model.

$$X = [T_1 \, T_2 \ldots T_m \, E'_1 \, E'_2 \cdots E'_n] = [X_1 \, X_2 \ldots X_k] \tag{29}$$

where k = m + n.

(3)　IGWO-BiLSTM model is established. In order to ensure that the model can train better results, assigning a smaller training set may lead to under-fitting the model, and assigning a larger training set may lead to over-fitting the model. The selected feature vectors are divided into training sets and test sets according to the ratio of 7:3. All data is normalized using the range of $[-1, 1]$ to avoid large differences between samples and improve the convergence speed of the model. The weight matrix and bias vector in the model are determined using the IGWO algorithm and substituted into the model for training. The trained lifespan prediction model is then tested with the test set, and the obtained prediction results, along with the remaining life data of the corresponding test set samples, are used to calculate the root mean square error (RMSE) and coefficient of determination (R2). R2 indicates the percentage of the model prediction array reaching the data itself, with a higher value indicating better regression accuracy. Evaluate the performance of the model using RMSE and R2 indicators. If the requirements are not met, the regression model can be reconstructed by modifying the parameters until the requirements are met. If the requirements are met, determine the IGWO-BiLSTM prediction model.

$$\text{RMS} = \sqrt{\frac{1}{N} \sum_{i=1}^{N} (\hat{y}_i - y_i)^2} \tag{30}$$

$$R^2 = \frac{\left(N \sum_{i=1}^{N} \hat{y}_i y_i - \sum_{i=1}^{N} \hat{y}_i \sum_{i=1}^{N} y_i\right)^2}{[N \sum_{i=1}^{N} \hat{y}_i^2 - (\sum_{i=1}^{N} \hat{y}_i)^2)][N \sum_{i=1}^{N} y_i^2 - (\sum_{i=1}^{N} y_i)^2]} \tag{31}$$

(4)　For signal data collected in real time, feature extraction is performed according to steps (1) and (2) before substituting the extracted feature vectors into the IGWO-BiLSTM prediction model established in step (3) to predict RUL in real time.

## 5. Life Prediction by Experimental Data

There are many failure modes of the ball screw. In order to compare and verify the model better, this paper analyzes and processes the life cycle signal of ball screw raceway wear failure. Because the wear failure of the inner ring and outer ring of the bearing has something in common with the wear failure of the ball screw raceway, the life-cycle signal of the bearing's outer ring failure is selected for simulation verification. This paper selected bearing life-cycle data sets provided by the University of Cincinnati. The experimental platform and structural diagram are shown in Figure 6:

On a single shaft, the bearing test equipment has four test bearings. An AC (alternating current) motor drives the shaft, which is connected by rubber belts. The rotation was maintained at a steady 2000 rpm. A spring mechanism adds a radial load of 6000 lbs. to the shaft and bearing. All of the bearings are greased by force. Both the flow and the temperature of the lubricant are controlled by an oil circulation system. Debris from the oil is collected by a magnetic plug that was inserted in the oil feedback channel as proof of bearing deterioration. When the magnetic plug's collected trash reaches a certain level and closes an electrical switch, the test will end. Four Rexnord ZA-2115 double-row bearings were installed on one shaft. Each row of bearings consists of 16 rollers with a pitch diameter of 2.815 inches, a roller diameter of 0.331 inches, and a conical contact angle of

15.171 inches. A PCB 353B33 high-sensitivity quartz ICPs (Integrated Circuit Piezoelectric) accelerometer was installed on each bearing box. Each bearing's outer race is equipped with four thermocouples for checking lubrication by the temperature reading of the bearing. A National Instruments DAQCard-6062E data acquisition card was used to record vibration data every 20 min. The data length is 20,480 points, and the data sample rate is 20 kHz. LabVIEW software from National Instruments was used to collect the data.

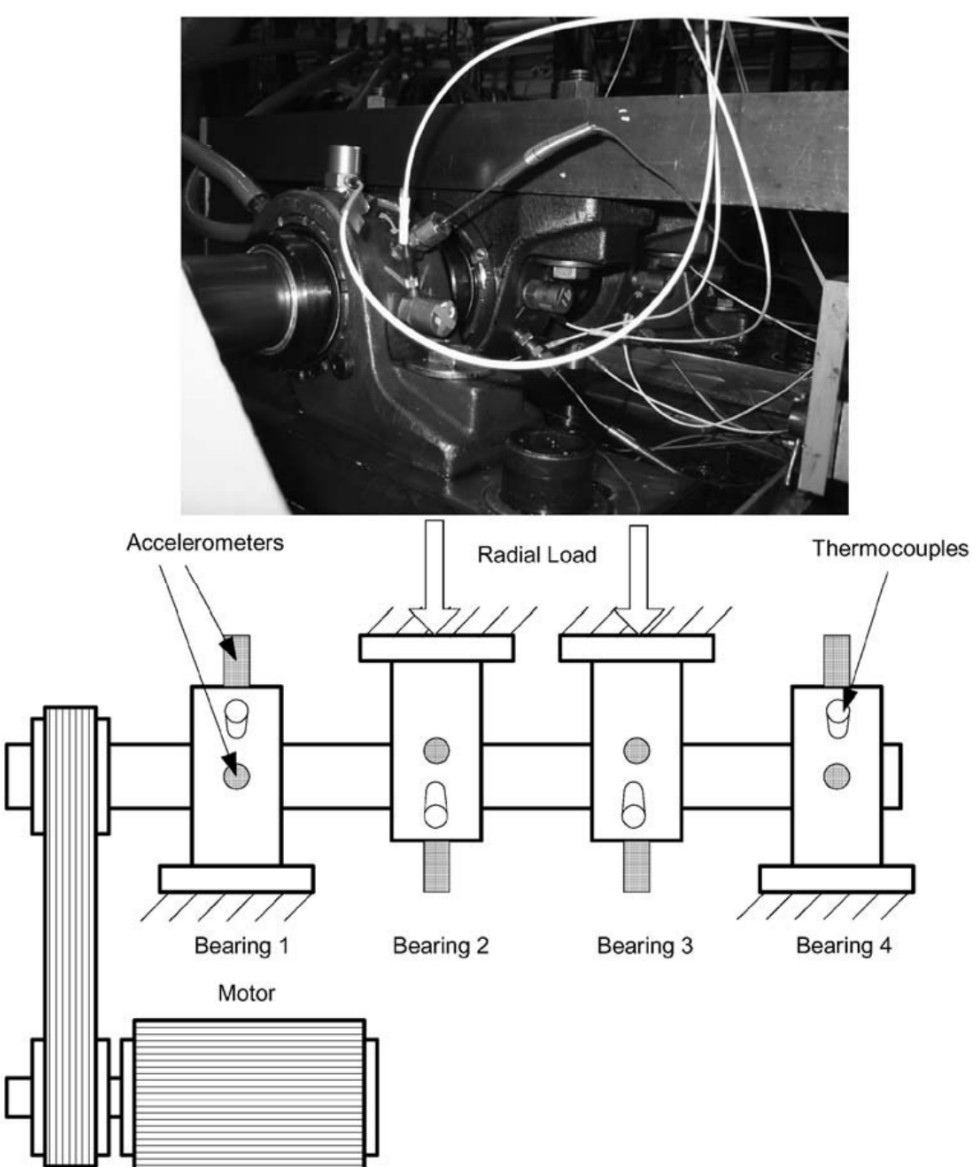

**Figure 6.** IMS experimental platform and structure sketch.

In this paper, the whole life-cycle vibration signal data of outer ring damage, inner ring damage, and roller damage were selected respectively. The time-domain image of the signal is shown in Figure 7. Due to the long-life cycle of the bearing, the signal fluctuation is relatively stable for a long time in the early stage, which is the healthy stage. In the later stage, due to the long-term wear of the bearing, the remaining life of the bearing is reduced, which causes the fluctuation amplitude of the vibration signal to change until the final failure.

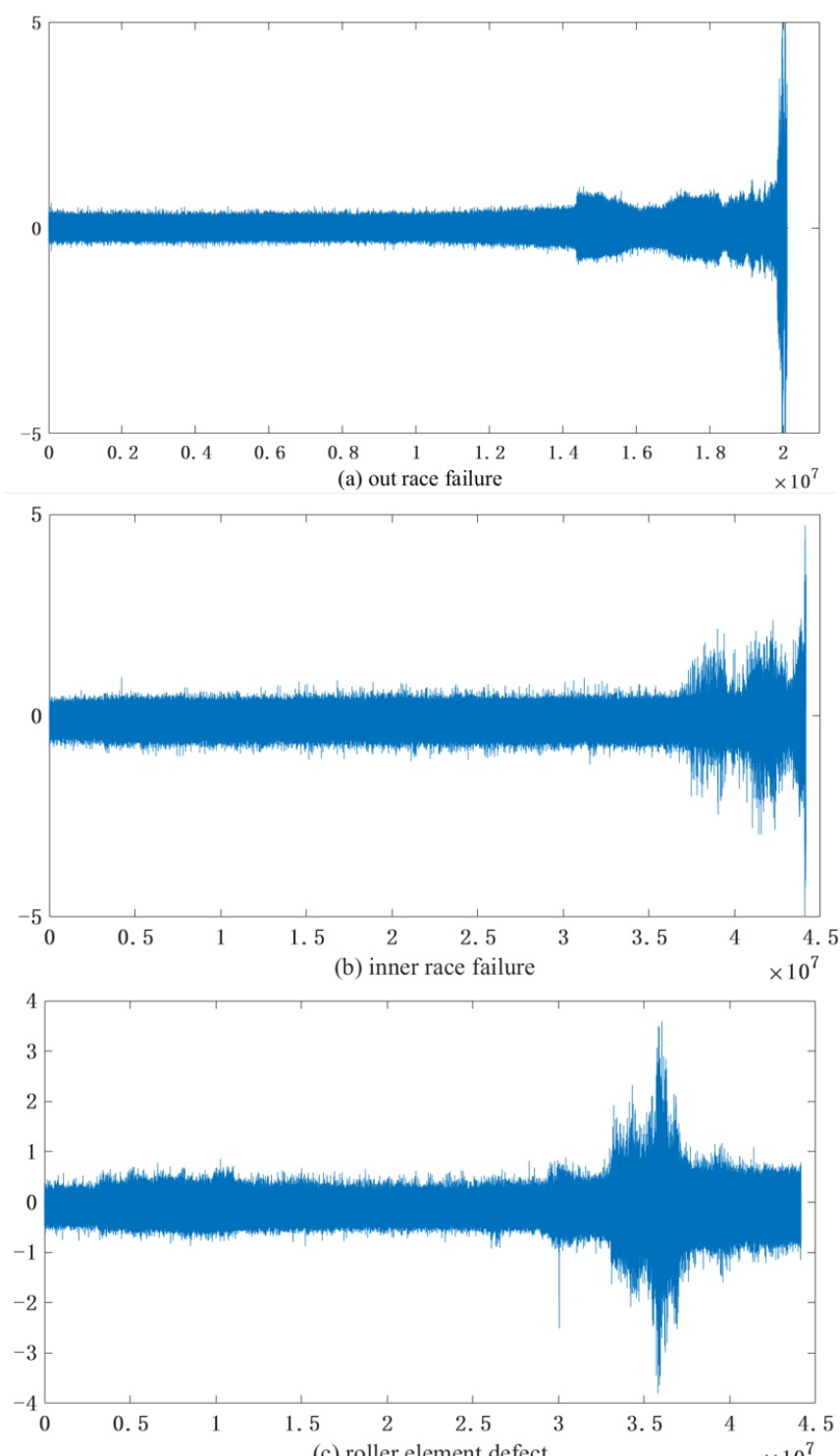

**Figure 7.** Whole life-cycle vibration signals under different failure modes of bearings.

This paper intercepted some signals from the entire life cycle as simulation data. The CEEMDAN method was utilized to decompose the residual life simulation signals at varying residual life levels, producing several IMF components. Figure 8 displays the original signal of the simulation data and the first 7 IMF components after decomposition at a residual life of 3200 h.

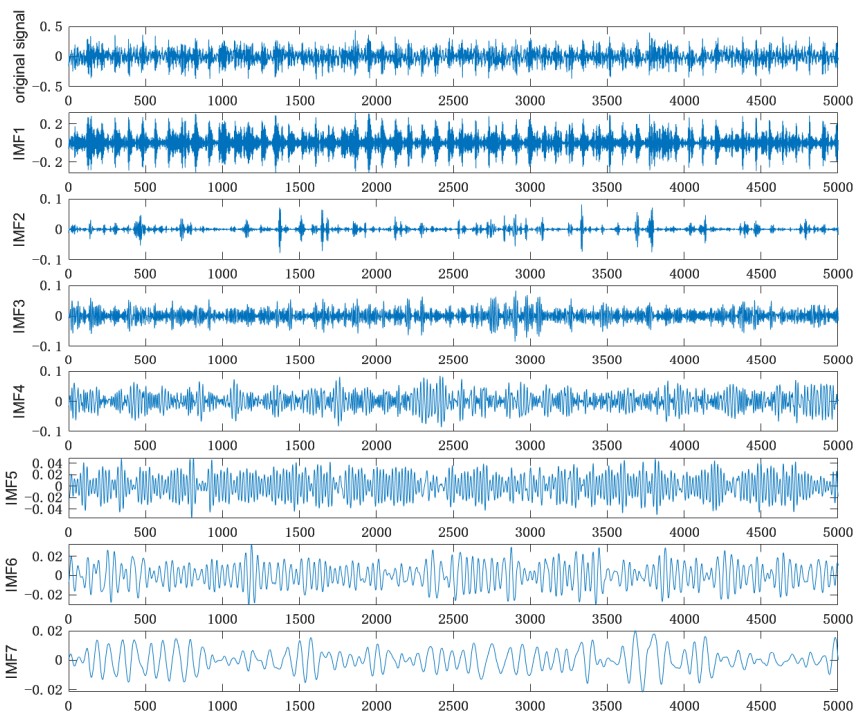

**Figure 8.** CEEMDAN decomposition of the first 7 components.

Based on Equation (6), the energy value of each component can be calculated, while substitution into Equation (28) allows for the calculation of the correlation coefficient between each component and the original signal. The six components with the highest correlation coefficient are selected as energy features in the degradation feature vector.

The time-domain feature T and the energy feature E′ are combined into a new degradation feature vector X (as shown in Table 2) as the model input features:

**Table 2.** Signal time domain, energy eigenvalues, and actual remaining life data.

| Serial Number | 1 | 2 | ... | 301 | 302 |
|---|---|---|---|---|---|
| Average amplitude | 0.080 | 0.080 | ... | 0.002 | 0.001 |
| Square root amplitude | 0.066 | 0.066 | ... | 0.002 | 0.001 |
| Standard deviation | 0.105 | 0.104 | ... | 0.001 | 0.001 |
| Root mean square | 0.145 | 0.104 | ... | 0.002 | 0.002 |
| $E_{IFM1}$ | 0.993 | 0.989 | ... | 0.993 | 0.988 |
| $E_{IFM2}$ | 0.005 | 0.006 | ... | 0.033 | 0.020 |
| $E_{IFM3}$ | 0.045 | 0.049 | ... | 0.078 | 0.131 |
| $E_{IFM4}$ | 0.094 | 0.114 | ... | 0.055 | 0.060 |
| $E_{IFM5}$ | 0.054 | 0.068 | ... | 0.047 | 0.038 |
| $E_{IFM6}$ | 0.021 | 0.020 | ... | 0.034 | 0.022 |
| Remaining life | 3200 | 3100 | ... | 20 | 10 |

IGWO was used to train the BiLSTM model to obtain the optimal model parameters. Using the optimal model parameters, the five models of standard LSTM, BiLSTM, BO-LSTM, BO-BiLSTM, and IGWO-LSTM were used to predict the RUL. The prediction results of each model are shown in Figure 9. The predicted points of the model are distributed on both sides of the real-value curve. The closer the scatter distance curve is, the more accurate the prediction is. Figure 9 shows the life prediction of the failure of the outer ring raceway by using six models. It can be seen from the diagram that the predicted scatter points of the IGWO-BiLSTM model are closer to the true value than the other five models.

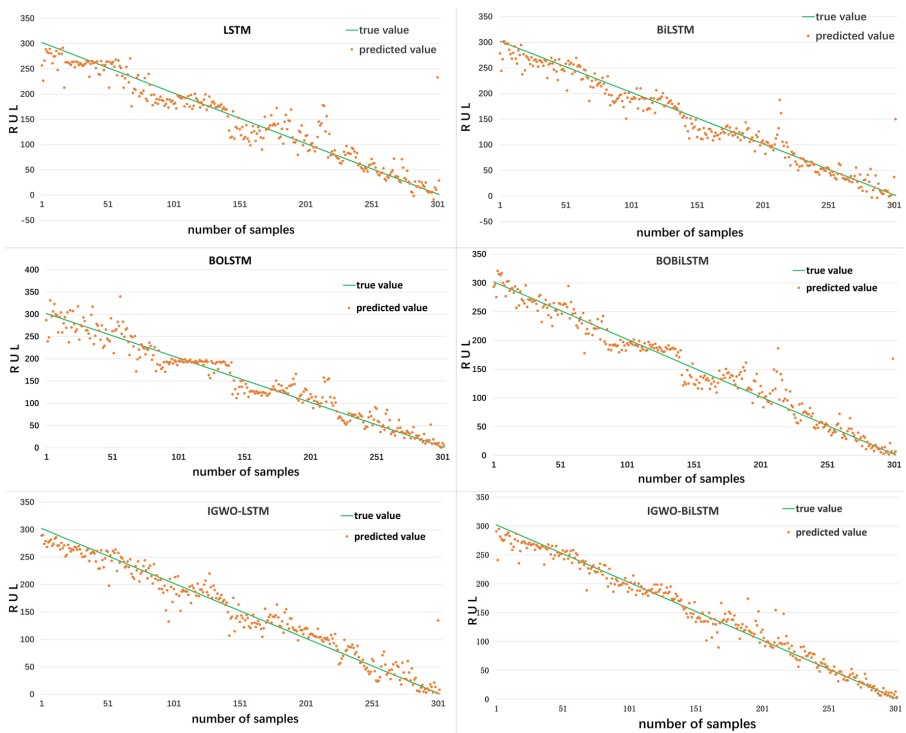

**Figure 9.** Outer ring raceway failure life prediction.

After using the full life-cycle vibration signal data as simulation data to predict the failure of the outer ring, in order to prove that the model is also applicable to inner ring raceway failure and roller failure, the study used the six models mentioned above to predict the life of the faults caused by these two types of failures. The predicted results are shown in Figures 10 and 11:

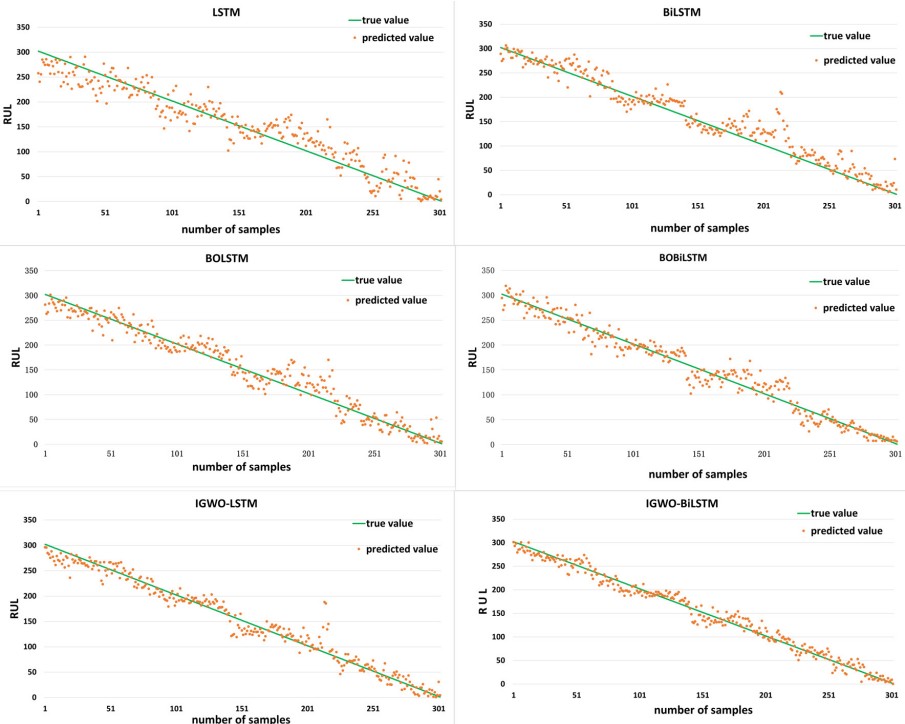

**Figure 10.** Inner ring raceway failure life prediction.

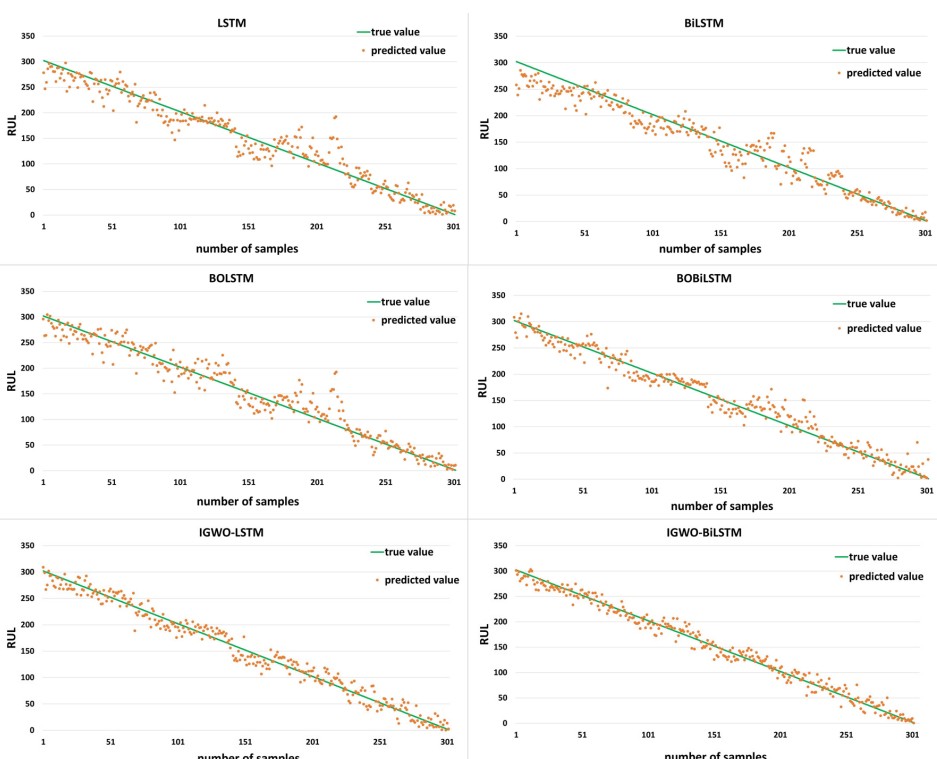

**Figure 11.** Roller failure life prediction.

In order to further verify the prediction accuracy and prediction ability of the model for the six different models, LSTM, BiLSTM, BO-LSTM, BO-BiLSTM, IGWO-LSTM and IGWO-BiLSTM, the two evaluation indicators, root mean square error (RMSE) and determination coefficient (R2), are selected to measure the prediction performance of the six models. RMSE and R2 are common indicators to measure the prediction performance of the model. RMSE is the square root of the average of the sum of squares of deviations between the predicted value and the actual value of the model. R2 refers to the proportion of the model prediction value that can explain the degree of variation of the actual value. Both RMSE and R2 are indicators for quantitative evaluation of model prediction accuracy, but they have different concerns. RMSE focuses on the difference between the predicted value and the actual value. The smaller the value, the more accurate the model prediction. The coefficient of determination focuses on the ability of the model to describe the actual data. The closer the value is to 1, the better the model can explain the data. Therefore, when evaluating the prediction performance of the model, the above two indicators can be considered comprehensively to ensure that the prediction accuracy and interpretation ability of the model are guaranteed at the same time. Figure 12 shows the prediction-accuracy analysis values of the RUL prediction results for different prediction models.

It can be clearly seen from the diagram that the root mean square error of the IGWO-BiLSTM regression model is smaller than that of the other five models when used to predict the remaining life of the screw. The coefficient of determination R2 predicted by the model is also the closest to 1. The minimum root mean square error indicates that the difference between the predicted value and the actual value of the method used in this paper is the smallest. When the coefficient of determination is 1, the predicted value of the model is the same as the actual value. The coefficient of determination of the predicted result of the IGWO-BiLSTM regression model is the closest to 1, the predicted value is the closest to the real value, and the model prediction is the most accurate. The results show that the IGWO-BiLSTM regression model proposed in this paper can accurately predict the remaining service life of ball screws.

For the real-time collected current signal data that performs the same process, the degradation feature vector is extracted according to the method described in this paper, and the remaining life of the screw corresponding to the current signal can be predicted by inputting the established IGWO-BiLSTM regression model.

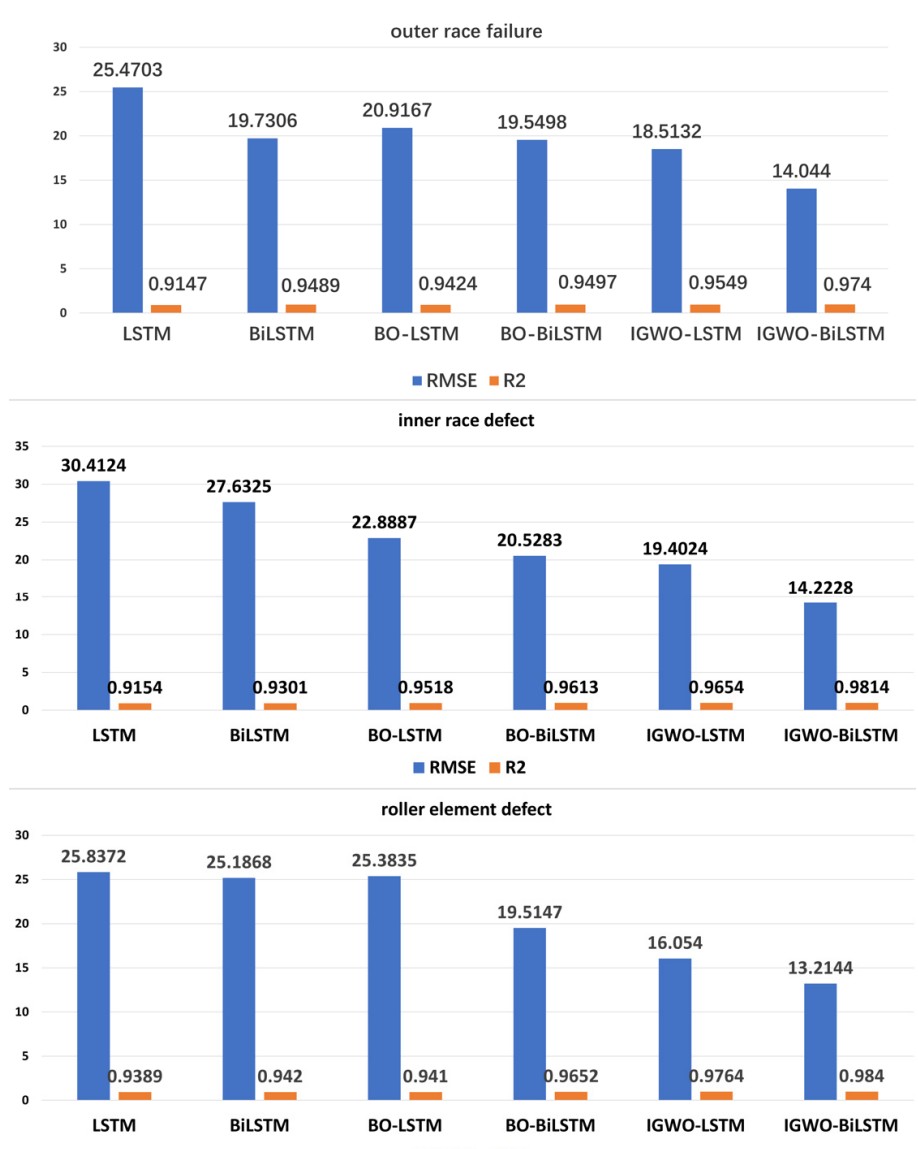

**Figure 12.** Comparison of model evaluation indicators.

## 6. Conclusions

For the data-driven ball screw remaining-life prediction problem, this paper mainly improves the traditional gray wolf optimization algorithm, improves the effect of algorithm optimization, and combines the improved algorithm with the BiLSTM neural network model to establish a new life-prediction regression model. Using the collected life-cycle signal data, the multivariate life-degradation feature vector is constructed. The regression model was validated and compared based on the actual remaining life data of a ball screw. The simulation results show that this method has better prediction accuracy and generalization ability compared to other models. This method is suitable for CNC machine tools to monitor and collect ball screw signals while predicting and evaluating the remaining life of the ball screw pair in real time based on the measured signals.

**Author Contributions:** Conceptualization, J.N. and Q.W.; methodology, J.N.; software, J.N. and Q.W.; validation, J.N., Q.W. and X.W.; formal analysis, J.N. and Q.W.; investigation, X.W.; resources, Q.W. and X.W.; data curation, Q.W. and X.W.; writing—original draft preparation, J.N.; writing—review and editing, J.N. and Q.W.; visualization, J.N. and X.W.; supervision, Q.W. and X.W.; project administration, Q.W.; funding acquisition, X.W. All authors have read and agreed to the published version of the manuscript.

**Funding:** This research received no external funding.

**Data Availability Statement:** Not applicable.

**Conflicts of Interest:** The authors declare no conflict of interest.

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
