# Peer review of "Study on Life Prediction Method of Ball Screw Base on Constructed Degradation Feature and IGWO-BiLSTM"

_actuators, doi:10.3390/act12060236_

Round 1

Reviewer 1 Report

In this paper, the IGWO-BiLSTM is developed for remaining life prediction of ball-screw system. In order to further improve the quality of this manuscript, the following comments should be carefully considered:

1.    What kind of fault condition does the paper consider? Single or multiple? Abrupt, incipient or intermittent?

2.    The theoretical contribution seems not strong, because only single fault condition is concerned. 

3.    The introduction part needs enhancement, more related works of feature extraction, e.g., “Intermittent fault diagnosis and prognosis for steer-by-wire system using composite degradation model,” IEEE Journal on Emerging and Selected Topics in Circuits and Systems, to be published, DOI: 10.1109/JETCAS.2023.3273207. and Intermittent fault modeling and RUL prediction for degraded electrical connectors in vibration environments, IEEE Transactions on Components Packaging and Manufacturing Technology, vol. 12, no. 5, pp. 769-777, 2022., should be discussed.

4.    The experimental platform should be clearly introduced.

5.    The future work should be mentioned.

The English Language should be improved due to the existence of typos. 

Reviewer 2 Report

The work presented is very interesting. However, there are some comments related to this work.

Abstract

·       The abstract does not provide a clear statement of the research question or problem that the paper addresses.

·       The abstract could be clearer about the specific contributions of the paper, such as how the proposed method addresses a specific gap in the literature.

·       It is recommended to define all abbreviations used in the paper in the introduction section, to ensure that readers can easily understand the terms and abbreviations used throughout the paper.

Introduction

·       Full words that define abbreviations should be capitalized and referred only once in the paper. For example, Intrinsic Mode Functions (IMFs).

·       The introduction should provide a brief overview of the topic being studied and the research question or problem that the study seeks to address. It should explain the significance of the research and its potential contributions to the field.

·       You should add another section ‘’Literature Review’’ in order to refer others’ work and methodologies in order to identify important gaps that the present approach deals with. It should be direct to the reader, at the end, what are the gaps and the key contributions of the present work in comparison to the identified gaps of other methods. Maybe you should add more papers regarding fault diagnosis methods in general. At the end of the paper, you could also compare your results with results from other papers. Below you can find some papers to add:

o   https://doi.org/10.1016/j.imu.2022.100875

o   https://doi.org/10.3390/s21030972

o   https://doi.org/10.1016/j.cie.2019.106024

o   https://doi.org/10.1016/j.autcon.2022.104695

o   https://doi.org/10.1016/j.energy.2023.126894

Lifetime prediction methods

·       ‘’All data is normalized using the range of [-1,1] to 318 avoid large differences between samples.’’ Please explain more about data normalization. Maybe you could also add a few references in order to support your choice.

·       Why the filtered feature vectors divided  into training and test sets in a 7:3 ratio?

·       You should add an ‘’Implementation’’ section in order to describe the software and hardware used for this study.

·       What are the model parameters?

Simulation data life prediction

 ·       Why did you choose for evaluation the metrics of RMAE and R2?

Conclusion

·       Next steps and future work should be explained in detail.

-

Reviewer 3 Report

The authors present a data driven prediction of remaining life in ball screws by filtering time and energy vectors by the use if the Pearson correlation coefficient. A model is presented pointing out a high accuracy in combination with a low RMSE.

The paper is expertly written, however suffering from 

a) overwhelming the reader with a flood of abbreviations being either not explained or being explained later in the text : this makes it undigestable over time

b) Table 1 is confusing as eight characteristics are tabled left and the others right. Better to write it downwards in 1 column

c) Line 192 /gray wolve is discredited in "optimal" (a-Wolve) and " best solution" (delta wolve), however does not explain what is meant

d) Authors et.al are sometimes Capital written sometimes normal (e.g. Line 48/54) : a choice according to the guidelines of MDPI should be taken.

e) The authors are spending apparently 2/3  of the paper in describing methods and solely 1/3 of presenting the results : this should be balanced according to  the paper guidelines

f) Only 1 Amplitude time signal is presented : in mechanical engineering it is common to represent a population failure by the use of Weibull statistics : this is missing and it should be explained.

g) In mechanical engineering Amplitude time recording is transferred by FFT into the frequency domain. An explanation why hasen't been done is missing.

h) It would be benefitial to explain the test rig in a more detailed manner,

The paper would benefit a lot by balancing the introduction with the experimental section, abbreviation of the description of the many methods, explaining the approach compared to standard FFT and the relation to Weibull statistics.

Apart from these critics, the paper  is excellent in competence !

The  paper suffers mainly from the flood of abbreviations and their explanations, either lacking or late in the text. This should be presented in a much clearer way.

Reviewer 4 Report

Good research has been done. In addition, the valuable results have been reported. The manuscript are written well and has a good structure. In summary, I believe that the present manuscript has a good potential to publish in the Journal. But, before publishing, it is necessary to provide the major revision. To this end, the authors should consider the following points:

1- The writing language is acceptable but need revision, so, it is strongly suggested to edit the final version of the article by a Native English Editor. 

2- Page 1 line 44, what is the meaning "18-dimensional degradation feature"? this sentence is not clear, please describe it more clear.

3- Page 2 line 61, the author name "WANG" should change to "Wang" and please this note in all over the article. 

4- The novelty of the present study compared to other research should be bold in the last paragraph of the introduction section. 

5- The literature review is not enough and should extend by using a recent published papers (after 2015). This section should be included the used method, results, advantages, and disadvantages to explain the innovation and the necessity of the present study. 

6-  Page 2 line 81, the phrase "Torres and colleagues" should change to "Torres et al.". 

7- It is strongly suggested to add Nomenclature in the article because there are several Abbreviation. 

8- It is better to show different steps of CEEMDAN algorithm as a schematic. 

9- Please check the equation (3), it seems that a parenthesis is missing.

10- Page 3 line 108, what is the meaning "resulting residue"?

11- Page 3 line 126, what is meaning "RNN"?

12- There are some symbols and parameters in equations 9-11 that do not explain in the  text. please do it. 

13- Related to Table 1, please add appropriate reference. 

14- Related to Figure 6, the numbers are not clear and please replace the image with high quality one. 

15- Figures 8-10 should be combined to one Figure and different parts. In addition, the numbers are not clear and please use the real scale and high quality one. 

16- Page 14 line 398-399, it is sated that the high accuracy technique was reported based on the root mean square value. But, it is necessary to interpret the results and describe the physical reasons. 

17- The conclusion should be rewrite including the most important achievements of the present research.  

The writing language is acceptable but need revision, so, it is strongly suggested to edit the final version of the article by a Native English Editor. 

Round 2

Reviewer 1 Report

The reviewer has no further comments.

Minor editing of English language is required.

Author Response

I have made a few modifications to the English grammar in the manuscript.

Reviewer 3 Report

Interesting paper presenting a novel methodology for remaining life assessment on structural parts. Due to the careful revision it is recommended for being published.

Author Response

Thank you very much for your suggestions to me, we will work harder in the future study.

Reviewer 4 Report

The authors tried to provide the revision manuscript based on the reviewers comments and also they response to the comments one by one. However, I believe that the changes are very large and the revised manuscript is okay compared to the initial submission. But, the below points should be considered before publication. 

1- Add Nomenclature in the article because there are several Abbreviation.

2- It is necessary to show different steps of CEEMDAN algorithm as a schematic.
